# Critical weaknesses in shielding strategies for COVID-19

**Cameron A. Smith[1], Christian A. Yates[1]☯, Ben Ashby[1,2]☯ ***

**1** Department of Mathematical Sciences, University of Bath, Bath, United Kingdom, **2** Milner Centre for Evolution, University of Bath, Bath, United Kingdom

☯ These authors contributed equally to this work.
* bna24@bath.ac.uk

**Data Availability Statement:** The MATLAB code for the implementation of the model can be found here: https://github.com/CameronSmith50/Critical-weaknesses-code.

## Abstract

The COVID-19 pandemic, caused by the coronavirus SARS-CoV-2, has led to a wide range of non-pharmaceutical interventions being implemented around the world to curb transmission. However, the economic and social costs of some of these measures, especially lockdowns, has been high. An alternative and widely discussed public health strategy for the COVID-19 pandemic would have been to 'shield' those most vulnerable to COVID-19 (minimising their contacts with others), while allowing infection to spread among lower risk individuals with the aim of reaching herd immunity. Here we retrospectively explore the effectiveness of this strategy using a stochastic SEIR framework, showing that even under the unrealistic assumption of perfect shielding, hospitals would have been rapidly overwhelmed with many avoidable deaths among lower risk individuals. Crucially, even a small (20%) reduction in the effectiveness of shielding would have likely led to a large increase (>150%) in the number of deaths compared to perfect shielding. Our findings demonstrate that shielding the vulnerable while allowing infections to spread among the wider population would not have been a viable public health strategy for COVID-19 and is unlikely to be effective for future pandemics.

## 1: Introduction

The COVID-19 pandemic has caused unprecedented health, economic, and societal challenges. As of February 2022, around 400 million cases and more than 5.5 million deaths have been confirmed, although the true numbers are thought to be far higher [1]. Prior to (and during) the rollout of vaccines, most countries introduced a range of non-pharmaceutical interventions (NPIs) to bring infections under control, including social distancing, travel restrictions, and lockdowns. While the effectiveness of different NPIs has varied within and between populations and over time, they have been largely effective at bringing outbreaks under control [2–4]. A widely discussed alternative approach would have been to limit most NPIs to the most vulnerable subpopulations while allowing those at lower risk to live with few or no restrictions [4–6]. 'Shielding' (or 'focused protection'), appeared to offer the possibility of avoiding the various societal costs of universal NPIs by leveraging the uneven risk profile of

**Funding:** BA is funded by Natural Environment Research Council grants NE/N014979/1 and NE/V003909/1. CAS is funded by Natural Environment Research Council grant NE/V003909/1. The funders had no role in study design, data collection and analysis, decision to publish, or preparation of the manuscript.

**Competing interests:** The authors have no competing interests to declare.

COVID-19, which is heavily skewed towards the elderly and those with certain pre-existing conditions [7, 8]. In theory, by allowing infections to spread with little to no suppression among the lower-risk population during a temporary shielding phase, the higher-risk population would subsequently be protected by herd immunity [9]. Several countries either openly or reportedly embraced this strategy during the early stages of the pandemic. Sweden, for example, chose to impose few restrictions on the general population while banning visits to long-term care (LTC) facilities [10], and the UK initially appeared to opt for a shielding strategy [11] before implementing a national lockdown. In the autumn of 2020, many countries experienced a resurgence in infections following the lifting of NPIs, leading to a renewed debate about the merits of shielding, driven by the Great Barrington Declaration which called for "focused protection of older people and other high-risk groups" while allowing uncontrolled viral transmission among lower-risk individuals [12, 13].

It is important to retrospectively assess the feasibility of shielding as a public health strategy, not only for public inquiries into COVID-19 and future pandemic preparedness, but also for countries where levels of vaccination remain low. Moreover, new variants may emerge which substantially escape vaccine-induced immunity, thus requiring a renewed choice between lockdowns and shielding while vaccines are updated. Although superficially appealing, serious practical and ethical concerns have been raised about shielding as a strategy to mitigate the impact of COVID-19 [14]. Yet there has been little mathematical modelling to determine the effectiveness of shielding under realistic conditions [4–6]. Crucially, the combined consequences of imperfect shielding, uneven distributions of immunity, and changes in contact behaviour among lower-risk individuals have yet to be explored.

Here, we use a mathematical model to evaluate whether shielding the most vulnerable while allowing infections to spread among lower-risk members of the population would have been an effective strategy to combat COVID-19. Our simulations are intended as illustrative examples of how shielding would have likely performed during the early stages of the pandemic, with the aim of informing future pandemic preparedness. We employ a stochastic SEIR model (see §2.1 and Fig 1) where the population is structured by risk of mortality (higher or lower risk) and location (community or LTC facilities). Our model is loosely based on an idealized large city in England (although our qualitative results would apply to similar countries) consisting of 1 million people, 7% of whom are at higher-risk of mortality from COVID-19, with 10% of higher-risk individuals situated in LTC facilities [15, 16]. We compare epidemics under no shielding, with imperfect (partial reduction in contacts for higher-risk individuals) and perfect shielding (no contacts for higher-risk individuals), with shielding restrictions lifted when cases fall below a given threshold (see §2.1).

This paper is arranged as follows. In §2 we introduce the modelling framework and the methods that we use: §2.1 contains the model formulation, §2.2 the calculation of the basic reproductive number for our model, and §2.3 the calculation of hospitalisation rates. In §3 we present our results, and we discuss our findings in §4.

## 2: Materials and methods

### 2.1: Model formulation

We simulate the spread of COVID-19 through the population of a large hypothetical city in England ($N = 1,000,000$). We consider a closed population (no births, non-disease related death or immigration) that is divided into three groups: a proportion $h$ of higher-risk individuals, with a proportion $ch$ of those living in the community ($H_C$) and $(1-c)h$ living in $n$ long-term care (LTC) facilities ($H_F^i$) (for $i = 1,...,n$), with the remaining fraction of the total population, $1-h$, being lower-risk individuals living in the community ($L$). We define the number of

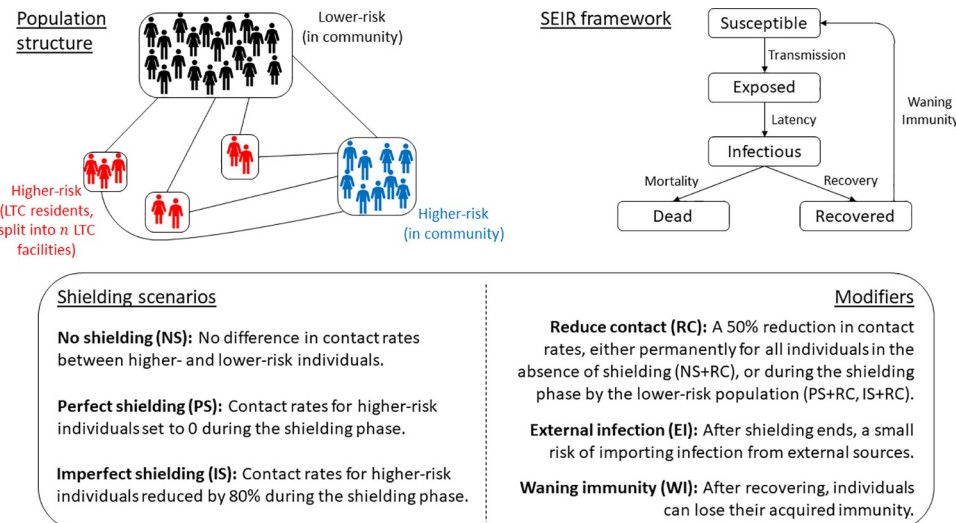

**Fig 1. Schematic for the model.** Top left: Population structure, with lines indicating contacts between subpopulations, each of which are well-mixed. Top right: Transitions through the different infection states in the SEIR model. Bottom: Description of the shielding scenarios and modifying assumptions.

people in each of the subpopulations to be $N_L$, $N_{H_C}$ and $N_{H_F^i}$ for the lower-risk community, higher-risk community, and long-term care residents in facility $i$ (for $i = 1,\ldots,n$), respectively. Using approximate figures for those classed as clinically extremely vulnerable (CEV) in England, we set $h = 0.07$ (7% at higher risk of mortality from COVID-19) and $c = 0.897$ (around 90% of higher-risk individuals live in the community). To reflect variation in the sizes of LTC facilities in England, we assume that LTC residents are distributed evenly over small, medium and large facilities: there are 120 small LTC facilities, each with 20 residents; 48 medium LTC facilities, each with 50 residents each, and 24 large LTC facilities, each of which house 100 residents. This gives an average of 37.5 residents per LTC facility, which is close to the UK average 39 [17].

We define the infection fatality ratio (IFR) for those at lower risk to be $\alpha_L$, and for those at higher risk to be $\alpha_H$, with ($\alpha_L < \alpha_H$). While many studies consider age stratified IFRs [7, 8, 18], there is comparatively little data on the IFR for CEV individuals. We assume that the lower-risk group consists of healthy people who are generally younger. IFR estimates for younger age groups range from 0.000097 [8] for the under 25s to 0.0052 [8] or 0.0094 [7] for the 45–64 age group. Conservatively, we choose a value towards the lower end of these estimates at 0.001 for the lower-risk group. For the higher-risk populations, we need to consider not only age, but also risk factors associated with being CEV. The majority of LTC residents are elderly, and so we look at the IFR for elderly populations as a proxy for this group. IFRs for 75-year-olds and over have been estimated to be as high as 0.1164 [7] and 0.147 [8], and previous modelling has assumed IFRs of 0.051 and 0.093 for the 70–79 and 80+ age groups, respectively [19]. We again choose a conservative estimate, setting the IFR for higher-risk individuals both in the community and in LTC facilities at 0.05. Our choice of IFRs are only approximations with the intention of illustrating how different shielding scenarios affect changes in cumulative deaths. Other reasonable choices of IFR for the different risk subcategories in our model do not qualitatively affect our conclusions. We set the average incubation period ($1/\sigma$) to be 5 days [19] and the average infectious period ($1/\Gamma$) to be 2 days [19], which are assumed to be the same for all individuals, and the basic reproduction number, $R_0$, to be 3 [20] (see §2 for derivation).

These parameters yield an unmitigated doubling time of around 3.2 days. For further simulations with $R_0 = 2.5$ and $R_0 = 3.5$, together with a sensitivity analysis for other key parameters, please see §A of the *S1 Text*. When included, we use a 240-day average period for waning immunity $(1/\nu)$, which is suggested to be an upper limit to the natural immunity conferred from SARS-CoV-2 according to the World Health Organisation [21].

Each individual in the population is assigned one of five epidemiological states: $S$ for those susceptible to the disease, $E$ for those exposed but not yet infectious, $I$ for those infected and able to transmit the disease, $R$ for those who have recovered (recovery is assumed to lead to full lifelong immunity), and $D$ for those who have died from the disease. We then define $S_i$, $E_i$, $I_i$, $R_i$ and $D_i$ for $i \in \{L, H_C, H_F^1, \ldots, H_F^n\}$ to be the total number of susceptible, exposed, infected, recovered, and dead individuals in subpopulation $i$. Susceptible individuals can become exposed through two pathways. Firstly, they may be "externally" infected from a member outside the population (for example, from another city or country), which we assume occurs at a rate $\eta_i(t)$ for subpopulation $i$ at time $t$. Alternatively, infected individuals of type $j$ may transmit the disease to susceptible individuals of type $i$ with rate $\beta_{ij} = \beta_0 r p_{ij}$, where $\beta_0$ is the transmission probability per contact, $r$ is the average number of contacts in the absence of restrictions, and $p_{ij}$ is the proportion of contacts that a person of type $i$ has with a person of type $j$. Written as a transmission matrix $\boldsymbol{\beta}$, we have:

$$\boldsymbol{\beta} = \beta_0 r \begin{pmatrix} p_{LL} & p_{LH_C} & p_{LH_F^1} & \cdots & p_{LH_F^n} \\ p_{H_C L} & p_{H_C H_C} & p_{H_C H_F^1} & \cdots & p_{H_C H_F^n} \\ p_{H_F^1 L} & p_{H_F^1 H_C} & p_{H_F^1 H_F^1} & \cdots & p_{H_F^1 H_F^n} \\ \vdots & \vdots & \vdots & \ddots & \vdots \\ p_{H_F^n L} & p_{H_F^n H_C} & p_{H_F^n H_F^1} & \cdots & p_{H_F^n H_F^n} \end{pmatrix}. \tag{1}$$

We assume that there is no direct contact between each of the LTC facilities, that a proportion $\lambda$ of an LTC resident's contacts occur within the same LTC facility, and that this proportion is the same for all LTC facilities. This yields:

$$p_{H_F^i H_F^j} = \lambda \delta_{i,j}, \tag{2}$$

where $\delta_{i,j}$ is the Kronecker delta, which takes the value 1 if $i = j$ and is 0 otherwise. The remainder of an LTC resident's contacts will occur with individuals in the community, normalised by the proportion of the population that is in the community:

$$p_{H_F^i L} = \frac{(1-h)(1-\lambda)}{1 - h(1-c)}, \tag{3A}$$

$$p_{H_F^i H_C} = \frac{ch(1-\lambda)}{1 - h(1-c)}, \tag{3B}$$

which holds for every $i \in \{1, \ldots, n\}$. The proportion of contacts that the lower- and higher-risk communities have with each care home is calculated as follows:

$$p_{LH_F^i} = \frac{N_{i+2}}{N} \frac{p_{H_F^i L}}{1-h}, \tag{4A}$$

$$p_{H_C H_F^i} = \frac{N_{i+2}}{N} \frac{p_{H_F^i H_C}}{hc}, \tag{4B}$$

where we have multiplied the contact rate in the opposite direction by the proportion of individuals that live in LTC facility $i$, and divide through by the proportion in each community group. We can calculate the remaining values in a similar way to $p_{H_F^i L}$ and $p_{H_F^i H_C}$ (i.e. by multiplying the remaining $1 - p_{H_F^i L}$ (and $1 - p_{H_F^i H_C}$) by the corresponding proportion of each subpopulation in the community), yielding the following transmission matrix:

$$\boldsymbol{\beta} = \beta_0 r \begin{pmatrix} \dfrac{(1-h)(1-\sum_i p_{LH_F^i})}{1-h(1-c)} & \dfrac{ch(1-\sum_i p_{LH_F^i})}{1-h(1-c)} & p_{LH_F^1} & \cdots & p_{LH_F^n} \\ \dfrac{(1-h)(1-\sum_i p_{H_C H_F^i})}{1-h(1-c)} & \dfrac{ch(1-\sum_i p_{H_C H_F^i})}{1-h(1-c)} & p_{H_C H_F^1} & \cdots & p_{H_C H_F^n} \\ p_{H_F^1 L} & p_{H_F^1 H_C} & \lambda & \cdots & 0 \\ \vdots & \vdots & \vdots & \ddots & \vdots \\ p_{H_F^n L} & p_{H_F^n H_C} & 0 & \cdots & \lambda \end{pmatrix}. \tag{5}$$

We implement non-pharmaceutical interventions (NPIs) by multiplying the transition matrix $\boldsymbol{\beta}$ element-wise by a shielding matrix $\boldsymbol{Q}$, whose entries lie between 0 and 1. A value of $Q_{ij} = 1$ denotes that interventions, if any, do not impact the contact rates between subpopulation $i$ and subpopulation $j$, so that contacts between the two occur as normal, while a value of $Q_{ij} = 0$ ceases all contacts between subpopulations $i$ and $j$. We further assume that the interventions are symmetric, so that $Q_{ij} = Q_{ji}$. The matrix $\boldsymbol{Q}$ is characterised by six different values and takes the following form:

$$\boldsymbol{Q} = \begin{pmatrix} q_1 & q_4 & q_5 & \cdots & q_5 \\ q_4 & q_2 & q_6 & \cdots & q_6 \\ q_5 & q_6 & q_3 & \cdots & 0 \\ \vdots & \vdots & \vdots & \ddots & \vdots \\ q_5 & q_6 & 0 & \cdots & q_3 \end{pmatrix}. \tag{6}$$

We consider three different shielding scenarios: no shielding (NS), imperfect shielding (IS), and perfect shielding (PS). In the imperfect and perfect shielding scenarios, shielding begins at the start of each simulation and ends once incidence falls below a threshold of 60 new cases per 100,000 in the population per week. This threshold is chosen based on the number of new cases recorded in the UK on the 1st April 2021 when shielding advice ended [22]. Once shielding ends it does not start again if cases rise. Coupled to each of these shielding scenarios, we include modifiers: reduced contact of the population during the shielding phase (RC) and the addition of an external force of infection into the community post shielding (EI). The RC modifier is motivated by Apple mobility data, which shows that in the week leading up to the first full lockdown in England on the 23rd March 2020, levels of movement may have dropped by around 70% [23], as measured by the number of requests for directions using Apple Maps. This indicates that members of the population may voluntarily reduce their contact when faced with an emerging pandemic. When there is no shielding strategy and the RC modifier is applied, we assume that all subpopulations reduce their contact equally. When either perfect or imperfect shielding is applied, the reduced contact is applied to the lower-risk population only as the higher-risk population already has reduced contact due to shielding. The second modifier, which introduces external infections, is motivated by those who enter the population and have the potential to infect those in the focal population. We assume that during the

shielding phase, the population is closed, and so this modifier is only applied after shielding ends. Table A in *S1 Text* shows values for the entries of $Q$ for the nine scenarios in §3.

To evolve the model system, we employ the Gillespie stochastic simulation algorithm (SSA) [24] (The code for our implementation can be found online [25]). The transition rates between the different states of the system are defined using the numbers of individuals in each of the subpopulations. Let $\underline{Y}i^t = (S_i, E_i, I_i, R_i, D_i)$ be the state variable for the $i^{\text{th}}$ subpopulation at time $t$. Then we can define the transition probabilities between states over a small time interval $(t, t+\delta t)$ to be as follows:

$$\mathbb{P}(\underline{Y}i^{t+\delta t} - \underline{Y}i^t = (-1, 1, 0, 0, 0)) = \left( \eta_i(t)S_i + \sum_j \frac{Q_{ij}\beta_{ij}S_iI_j}{N_j} \right)\delta t, \tag{7A}$$

$$\mathbb{P}(\underline{Y}i^{t+\delta t} - \underline{Y}i^t = (0, -1, 1, 0, 0)) = \sigma E_i \delta t, \tag{7B}$$

$$\mathbb{P}(\underline{Y}i^{t+\delta t} - \underline{Y}i^t = (0, 0, -1, 1, 0)) = \Gamma I_i(1 - \alpha_i)\delta t, \tag{7C}$$

$$\mathbb{P}(\underline{Y}i^{t+\delta t} - \underline{Y}i^t = (0, 0, -1, 0, 1)) = \Gamma I_i\alpha_i\delta t, \tag{7D}$$

$$\mathbb{P}(\underline{Y}i^{t+\delta t} - \underline{Y}i^t = (1, 0, 0, -1, 0)) = v\delta t, \tag{7E}$$

which holds for every subpopulation $i$, whilst holding each of the other $\underline{Y}j^t$'s constant for every $j \neq i$. Each of the above probabilities is associated with the transition of an individual between successive disease states. The first is the conversion of a susceptible to an exposed individual through coming into contact with an infected individual from any of the other subpopulations $j \in \{L, H_C, H_F^1, \ldots, H_F^n\}$. Note that the first term in the bracket is the external infection term, which is always 0 during the shielding phase, and when included, is non-zero only in the lower-risk and higher-risk subpopulations in the community. The second characterises the transition from being exposed to infectious, the third the recovery of an infected individual and the fourth the death of an infected individual. We run each of our stochastic simulations for 600 days and calculate averages and variances over 100 independent repeats, initialised with 10 lower-risk individuals in the infected class. We remove any instances of immediate stochastic die out from our analysis (this is a rare occurrence).

## 2.2: The basic reproduction number

To approximate the basic reproduction number for our simulation, we employ the mean-field ordinary differential equations (ODEs) for each subpopulation, for which we combine all LTC facility residents into one large subpopulation. In this section, we assume that the elements $\beta_{ij}$ contain the appropriate shielding matrix term. The mean-field equations (assuming no interventions) are:

$$\frac{dS_L}{dt} = -\frac{\beta_{LL}S_LI_L}{N_L} - \frac{\beta_{LH_C}S_LI_{H_C}}{N_{H_C}} - \frac{\beta_{LH_F}S_LI_{H_F}}{N_{H_F}}, \tag{8A}$$

$$\frac{dS_{H_C}}{dt} = -\frac{\beta_{H_CL}S_{H_C}I_L}{N_L} - \frac{\beta_{H_CH_C}S_{H_C}I_{H_C}}{N_{H_C}} - \frac{\beta_{H_CH_F}S_{H_C}I_{H_F}}{N_{H_F}}, \tag{8B}$$

$$\frac{dS_{H_F}}{dt} = -\frac{\beta_{H_F L} S_{H_F} I_L}{N_L} - \frac{\beta_{H_F H_C} S_{H_F} I_{H_C}}{N_{H_C}} - \frac{\beta_{H_F H_F} S_{H_F} I_{H_F}}{N_{H_F}}, \tag{8C}$$

$$\frac{dE_L}{dt} = \frac{\beta_{LL} S_L I_L}{N_L} + \frac{\beta_{LH_C} S_L I_{H_C}}{N_{H_C}} + \frac{\beta_{LH_F} S_L I_{H_F}}{N_{H_F}} - \sigma E_L, \tag{8D}$$

$$\frac{dE_{H_C}}{dt} = \frac{\beta_{H_C L} S_{H_C} I_L}{N_L} + \frac{\beta_{H_C H_C} S_{H_C} I_{H_C}}{N_{H_C}} + \frac{\beta_{H_C H_F} S_{H_C} I_{H_F}}{N_{H_F}} - \sigma E_{H_C}, \tag{8E}$$

$$\frac{dE_{H_F}}{dt} = \frac{\beta_{H_F L} S_{H_F} I_L}{N_L} + \frac{\beta_{H_F H_C} S_{H_F} I_{H_C}}{N_{H_C}} + \frac{\beta_{H_F H_F} S_{H_F} I_{H_F}}{N_{H_F}} - \sigma E_{H_F}, \tag{8F}$$

$$\frac{dI_L}{dt} = \sigma E_L - \Gamma I_L, \tag{8G}$$

$$\frac{dI_{H_C}}{dt} = \sigma E_{H_C} - \Gamma I_{H_C}, \tag{8H}$$

$$\frac{dI_{H_F}}{dt} = \sigma E_{H_F} - \Gamma I_{H_F}. \tag{8I}$$

The recovered and death classes have been omitted here because they are not required for the calculation. We employ the next generation matrix method [26] in order to find the basic reproduction number. We linearise the infected state ODEs ($E_i$ and $I_i$) in system (8) about the disease-free equilibrium

$$\underline{S}_0 = (S_L, S_{H_C}, S_{H_F}, E_L, E_{H_C}, E_{H_F}, I_L, I_{H_C}, I_{H_F}) = (N_L, N_{H_C}, N_{H_F}, 0, 0, 0, 0, 0, 0), \tag{9}$$

by writing $\underline{x} = (E_L, E_{H_C}, E_{H_F}, I_L, I_{H_C}, I_{H_F})^T$ (where the superscript $T$ denotes the transpose) and obtaining an ODE $\dot{\underline{x}} = A\underline{x}$, where:

$$A = \begin{pmatrix} -\sigma & 0 & 0 & \beta_{LL} & \beta_{LH_C}\dfrac{N_L}{N_{H_C}} & \beta_{LH_F}\dfrac{N_L}{N_{H_F}} \\ 0 & -\sigma & 0 & \beta_{H_C L}\dfrac{N_{H_C}}{N_L} & \beta_{H_C H_C} & \beta_{H_C H_F}\dfrac{N_{H_C}}{N_{H_F}} \\ 0 & 0 & -\sigma & \beta_{H_F L}\dfrac{N_{H_F}}{N_L} & \beta_{H_F H_C}\dfrac{N_{H_F}}{N_{H_C}} & \beta_{H_F H_F} \\ \sigma & 0 & 0 & -\Gamma & 0 & 0 \\ 0 & \sigma & 0 & 0 & -\Gamma & 0 \\ 0 & 0 & \sigma & 0 & 0 & -\Gamma \end{pmatrix}. \tag{10}$$

We split matrix (10) into components $T$ and $\Sigma$ which contain the transmission terms (or the terms relating to the mechanism by which individuals enter this truncated system) and all

other terms respectively, so that:

$$T = \begin{pmatrix} 0 & 0 & 0 & \beta_{LL} & \beta_{LH_C}\dfrac{N_L}{N_{H_C}} & \beta_{LH_F}\dfrac{N_L}{N_{H_F}} \\ 0 & 0 & 0 & \beta_{H_CL}\dfrac{N_{H_C}}{N_L} & \beta_{H_CH_C} & \beta_{H_CH_F}\dfrac{N_{H_C}}{N_{H_F}} \\ 0 & 0 & 0 & \beta_{H_FL}\dfrac{N_{H_F}}{N_L} & \beta_{H_FH_C}\dfrac{N_{H_F}}{N_{H_C}} & \beta_{H_FH_F} \\ 0 & 0 & 0 & 0 & 0 & 0 \\ 0 & 0 & 0 & 0 & 0 & 0 \\ 0 & 0 & 0 & 0 & 0 & 0 \end{pmatrix}, \quad (11)$$

$$\Sigma = \begin{pmatrix} -\sigma & 0 & 0 & 0 & 0 & 0 \\ 0 & -\sigma & 0 & 0 & 0 & 0 \\ 0 & 0 & -\sigma & 0 & 0 & 0 \\ \sigma & 0 & 0 & -\Gamma & 0 & 0 \\ 0 & \sigma & 0 & 0 & -\Gamma & 0 \\ 0 & 0 & \sigma & 0 & 0 & -\Gamma \end{pmatrix}. \quad (12)$$

The next generation matrix $K$ is then given by $K = -T\Sigma^{-1}$:

$$K = -T\Sigma^{-1}$$
$$= \begin{pmatrix} \dfrac{\beta_{LL}}{\Gamma} & \dfrac{\beta_{LH_C}}{\Gamma}\dfrac{N_L}{N_{H_C}} & \dfrac{\beta_{LH_F}}{\Gamma}\dfrac{N_L}{N_{H_F}} & \dfrac{\beta_{LL}}{\Gamma} & \dfrac{\beta_{LH_C}}{\Gamma}\dfrac{N_L}{N_{H_C}} & \dfrac{\beta_{LH_F}}{\Gamma}\dfrac{N_L}{N_{H_F}} \\ \dfrac{\beta_{H_CL}}{\Gamma}\dfrac{N_{H_C}}{N_L} & \dfrac{\beta_{H_CH_C}}{\Gamma} & \dfrac{\beta_{H_CH_F}}{\Gamma}\dfrac{N_{H_C}}{N_{H_F}} & \dfrac{\beta_{H_CL}}{\Gamma}\dfrac{N_{H_C}}{N_L} & \dfrac{\beta_{H_CH_C}}{\Gamma} & \dfrac{\beta_{H_CH_F}}{\Gamma}\dfrac{N_{H_C}}{N_{H_F}} \\ \dfrac{\beta_{H_FL}}{\Gamma}\dfrac{N_{H_F}}{N_L} & \dfrac{\beta_{H_FH_C}}{\Gamma}\dfrac{N_{H_F}}{N_{H_C}} & \dfrac{\beta_{H_FH_F}}{\Gamma} & \dfrac{\beta_{H_FL}}{\Gamma}\dfrac{N_{H_F}}{N_L} & \dfrac{\beta_{H_FH_C}}{\Gamma}\dfrac{N_{H_F}}{N_{H_C}} & \dfrac{\beta_{H_FH_F}}{\Gamma} \\ 0 & 0 & 0 & 0 & 0 & 0 \\ 0 & 0 & 0 & 0 & 0 & 0 \\ 0 & 0 & 0 & 0 & 0 & 0 \end{pmatrix}. \quad (13)$$

The basic reproduction number, $R_0$, is given by the leading eigenvalue of $K$. To simplify the resulting characteristic polynomial, we note that the matrix $\boldsymbol{\beta}$ takes the following form, calculated in an analogous way to §2.1 above:

$$\boldsymbol{\beta} = \begin{pmatrix} \beta_{LL} & \beta_{LH_C} & \beta_{LH_F} \\ \beta_{H_CL} & \beta_{H_CH_C} & \beta_{H_CH_F} \\ \beta_{H_FL} & \beta_{H_FH_C} & \beta_{H_FH_F} \end{pmatrix} = \begin{pmatrix} b_1q_1 & b_2q_4 & b_3q_5 \\ b_1q_4 & b_2q_2 & b_3q_6 \\ b_4q_5 & b_5q_6 & b_6q_3 \end{pmatrix}, \quad (14)$$

where:

$$b_1 = \beta_0 r \frac{(1-h)(1-b_3)}{1-h(1-c)}, \tag{15A}$$

$$b_2 = \beta_0 r \frac{ch(1-b_3)}{1-h(1-c)}, \tag{15B}$$

$$b_3 = \beta_0 r \frac{h(1-c)(1-\lambda)}{1-h(1-c)}, \tag{15C}$$

$$b_4 = \beta_0 r \frac{(1-h)(1-\lambda)}{1-h(1-c)}, \tag{15D}$$

$$b_5 = \beta_0 r \frac{ch(1-\lambda)}{1-h(1-c)}, \tag{15E}$$

$$b_6 = \beta_0 r \lambda, \tag{15F}$$

and the $q_i$s are as in Eq (6). Substituting Eqs (15) into (13) and simplifying, we find the characteristic polynomial, $P(s)$, to be:

$$P(s) = s^3 \left[ s^3 - \frac{(b_1 q_1 + b_2 q_2 + b_6 q_3)}{\Gamma} s^2 \right.$$
$$+ \frac{b_6 q_3 (b_1 q_1 + b_2 q_2) - b_3 (b_4 q_5^2 + b_5 q_6^2) + b_1 b_2 (q_1 q_2 - q_4^2)}{\Gamma^2} s$$
$$\left. - \frac{b_1 b_2 b_6 q_3 (q_1 q_2 - q_3 q_4) + b_1 b_3 b_5 q_6 (q_4 q_5 - q_1 q_6) + b_2 b_3 b_4 q_5 (q_4 q_6 - q_2 q_5)}{\Gamma^3} \right]. \tag{16A}$$

Under the no shielding scenario, $q_i = 1$ for every $i \in \{1,\dots,6\}$ and we obtain the simplified characteristic polynomial:

$$P(s) = s^4 \left[ s^2 - \frac{1}{\Gamma} (b_1 + b_2 + b_6) s + \frac{1}{\Gamma^2} (b_6 (b_1 + b_2) - b_3 (b_4 + b_5)) \right]. \tag{16B}$$

Employing the fact that $b_1 + b_2 + b_3 = b_4 + b_5 + b_6 = \beta_0 r$, we obtain the form:

$$P(s) = s^4 \left( s - \frac{\beta_0 r}{\Gamma} \right) \left( s - \frac{b_1 + b_2 + b_6 - \beta_0 r}{\Gamma} \right). \tag{17}$$

The largest of the two eigenvalues that result from setting the characteristic polynomial to zero is $\beta_0 r / \Gamma$, and hence:

$$R_0 = \frac{\beta_0 r}{\Gamma}. \tag{18}$$

## 2.3: Hospitalisation

To calculate the impact of our interventions on the occupancy of intensive care units (ICUs) during the epidemic, we utilise data from [19] on the age distribution of patients requiring ICU treatment (Table C in *S1 Text*). These data allow us to calculate the probability that an individual who is infected requires ICU treatment. This probability is then *a posteriori* applied

to our infection curve to estimate the numbers of people who would be in ICU. To calculate the probability of requiring ICU treatment given that an individual is symptomatic in the lower-risk group (assumed to be 66% of our infected class), we take a weighted average over all ages up to and including 64 (in a similar way to the calculation of IFR values), while the higher-risk subpopulations use ages 65 and over as a proxy. This yields probabilities of requiring ICU treatment for the lower- and higher-risk (community and LTC facilities) subpopulations, $\pi_L$ and $\pi_H$, of:

$$\pi_L = \sum_{\ell \in \mathcal{A}_L} pop_\ell \times ICU_\ell,$$

$$\pi_H = \sum_{\ell \in \mathcal{A}_H} pop_\ell \times ICU_\ell,$$

where $\mathcal{A}_L$ is the set of age categories below the age of 64, and $\mathcal{A}_H$ is the set of age categories above the age of 65. Also, $pop_\ell$ is the proportion of the population in the age category $\ell$, and $ICU_\ell$ is the probability that a symptomatic individual in age category $\ell$ is admitted to ICU. Each individual admitted to ICU is assumed to stay for ten days (on average) [19]. The ICU capacity for the UK is approximately 8 ICU beds per 100,000 people [19].

## 3: Results

An unmitigated epidemic with no shielding (NS) would have represented the worst-case scenario (Fig 2, col. 1), with an estimated peak incidence of 4149.0±274.1 (mean ± standard deviation) cases per 100,000 and a total of 415.1±6.5 deaths per 100,000, equivalent to 230,795±3,615 total deaths in England. This is likely a conservative estimate, as hospitals would have been rapidly overwhelmed, with intensive care unit (ICU) capacity exceeded by a factor of approximately 18 at the peak of the epidemic (Fig 2D). In contrast, perfect shielding (PS) would have been the best-case scenario (although unattainable) (Fig 2, col. 3), with a peak incidence of 3470.5±456.1 cases per 100,000 but only 87.6±3.4 deaths per 100,000. Perfect shielding represents a substantial improvement on an unmitigated epidemic (79% reduction in deaths), but almost all deaths would have been among lower-risk members of the population. In England, this would have equated to nearly 50,000 deaths among lower-risk individuals. As in the no shielding scenario, this is likely a conservative estimate as hospital capacity would have been rapidly overwhelmed: assuming an average duration of treatment of 10 days, ICU bed capacity in England would have been exceeded by over a factor of 10 at the peak of the epidemic with perfect shielding (see §2.3).

However, shielding would have been impossible to implement perfectly. LTC residents, for example, have contact with staff, and many higher-risk individuals in the community live with or receive care from lower-risk individuals. Between 14 May and 16 July 2020, only 58–63% of CEV people in England were able to follow guidelines to avoid contact completely [16], and despite strict restrictions on LTC facilities in Sweden and England during the first wave of the pandemic, a high proportion of COVID-19 deaths were LTC residents [15]. Imperfect shielding, the first critical weakness of this strategy, therefore represents a more realistic scenario. If shielding had been only 80% effective while an otherwise unmitigated epidemic spread through the lower-risk population, we estimate that there would have been large outbreaks among higher-risk individuals both in the community and in LTC facilities (Fig 2, col. 2) leading to a much higher death rate of 221.7±3.8 per 100,000. Even a relatively small reduction in shielding effectiveness (20%) would have therefore led to a sharp increase in deaths (>150%) compared to perfect shielding (Fig 3). Higher-risk individuals in the community would have been disproportionately affected due to imperfect shielding, with 200% higher death rates

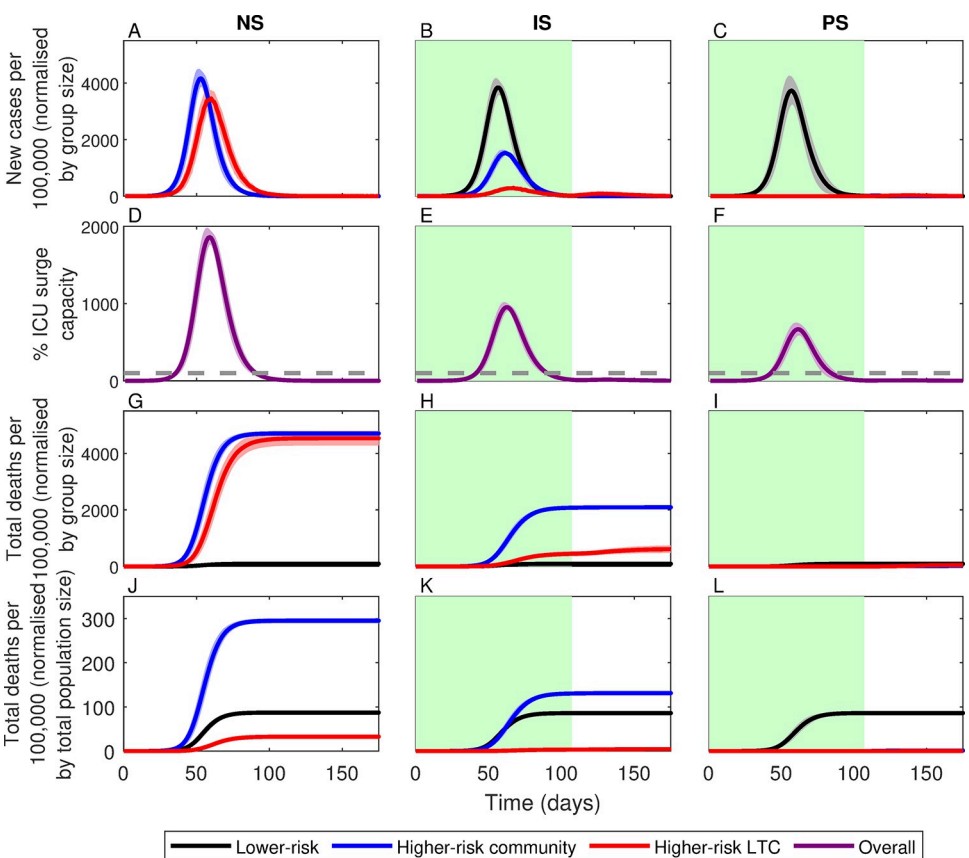

**Fig 2. Simulations of no (NS), imperfect (IS) and perfect (PS) shielding.** Lines correspond to means for groups at: lower-risk (black), higher-risk in the community (blue) and in LTC facilities (red), with shading indicating ± 1 SD. Green shading indicates the shielding phase. Top row: daily number of new cases per 100,000 members of each group (i.e., new cases in each group multiplied by 100,000 and divided by group size). Second row: percentage of surge capacity ICU beds in demand (horizontal dashed line indicates full capacity). Third row: cumulative number of deaths per 100,000 members of each group (i.e., total deaths in each group multiplied by 100,000 and divided by group size). Bottom row: cumulative number of deaths per 100,000 members of the total population (i.e., total deaths multiplied by 100,000 and divided by total population size). Data averaged over 100 identically initialized stochastic repeats (see *Materials and methods*).

compared to LTC residents. Again, these figures are likely to be conservative as we estimate that hospital capacity would have been exceeded by a factor of 9.5 at the peak of the epidemic.

The second critical weakness of the shielding strategy is that it relies on large numbers of lower-risk individuals becoming infected to build up immunity in the population. Yet many people would have likely changed their behaviour to avoid infection, leading to smaller, longer outbreaks with fewer infections and potentially leaving immunity levels below the threshold needed to prevent subsequent outbreaks [9]. Prior to England's first national lockdown, mobility data shows that movement dropped by as much as 70% [23], and many people continued to take precautions, such as mask wearing and working from home, even after restrictions were fully lifted in July 2021 [27]. A resurgence in cases leading to a second, deadlier wave, occurs in our modelling when reduced contact (50%) among lower risk individuals is combined with shielding, whether imperfect (IS+RC, 321.2±11.5 deaths per 100,000) or perfect (PS +RC, 299.5±7.5 deaths per 100,000) (Fig 4, cols 2–3). Reduced contact among lower-risk individuals leads to much smaller peaks in incidence and hospitalizations, although ICU surge

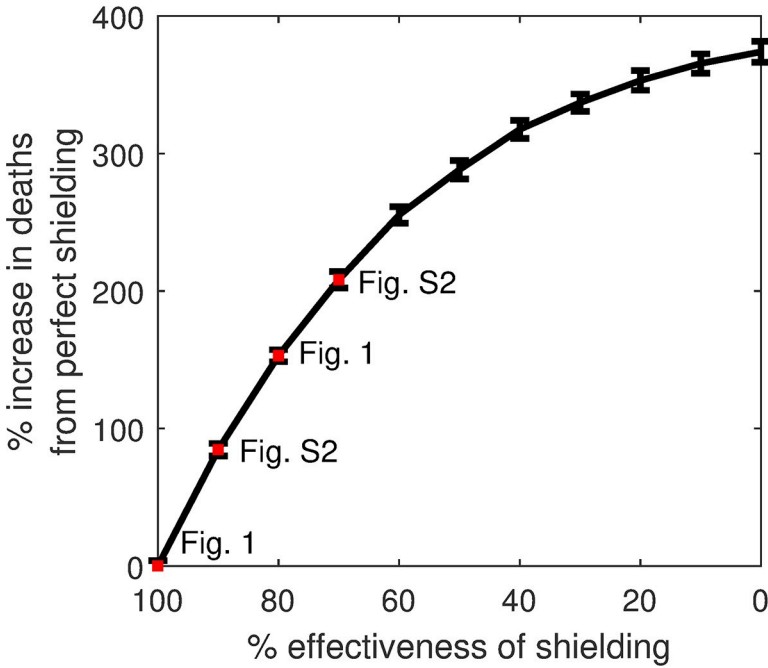

**Fig 3. Relative deaths under varying levels of imperfect shielding (compared to perfect shielding).** Red dots and labels correspond to figures showing these scenarios, and vertical bars indicate ± 1 standard deviation.

capacity would still likely have been exceeded without further restrictions (Fig 4). It is also likely that people would have increasingly limited their contacts if healthcare services were overwhelmed, which would have further reduced the likelihood of reaching the herd immunity threshold before shielding ended.

A third critical weakness of the shielding strategy is that herd immunity only confers indirect protection and is only temporary. In theory, herd immunity would have been achieved primarily through infection of lower risk members of the population, conferring indirect protection to higher-risk individuals by preventing large outbreaks following the lifting of restrictions (See §C and Fig P in *S1 Text*). Yet many vulnerable members of the population would have remained at risk of infection after shielding had ended, from residual transmission in the community, from externally imported (EI) infections (e.g., due to international travel; Fig 5) or from a resurgence in community transmission due to waning immunity (Fig 6). Crucially, a heterogeneous distribution of immunity would have arisen in the population during the shielding phase, with LTC facilities remaining highly susceptible to local outbreaks once restrictions were lifted. If the shielding phase were to end prematurely while community transmission was still high or if infections were imported from other areas, local outbreaks would have likely still occurred in LTC facilities even if the population as a whole was above the herd immunity threshold. Similar effects have been observed for other pathogens, notably measles outbreaks in communities with low vaccination rates [28]. In our simulations, we see that an external force of infection leads to a steady increase in deaths after shielding is lifted (Fig 5), both among higher-risk individuals in the community despite herd-immunity being achieved, and among clusters of higher-risk individuals in LTC facilities in which herd immunity has not been achieved locally. Similarly, waning immunity leads to a resurgence in cases following the relaxation of shielding, leading to a substantial increase in deaths among those at greatest risk (Fig 6).

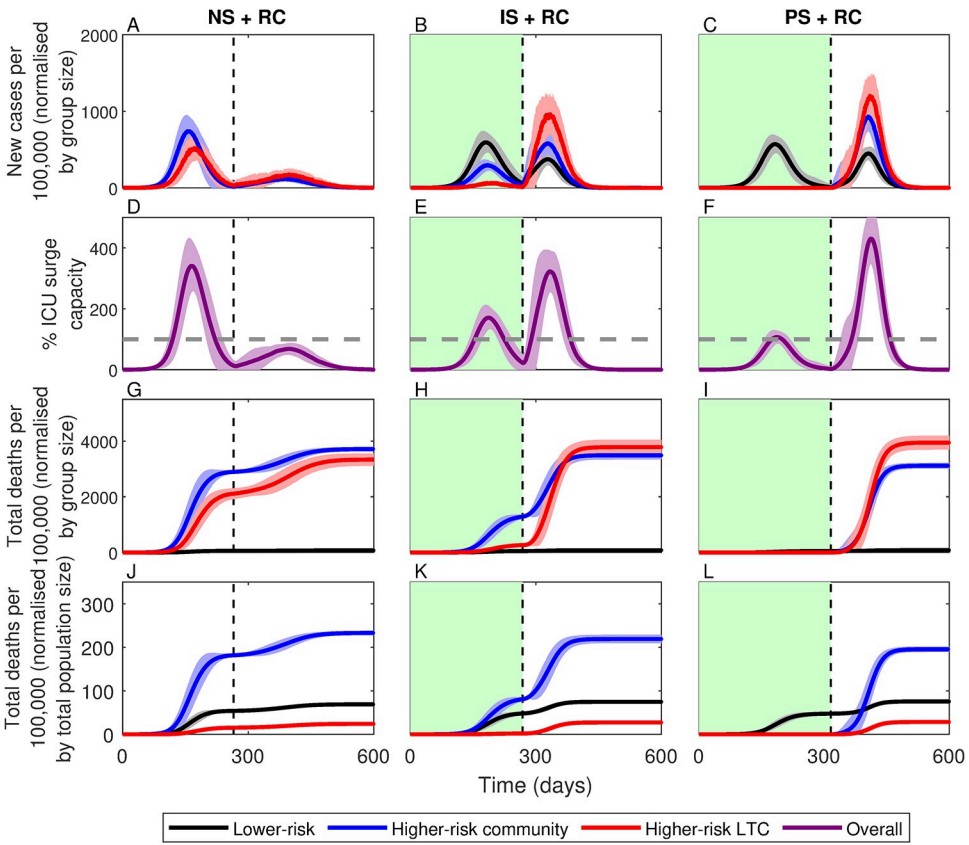

**Fig 4. Simulations of the three shielding scenarios with 50% reduced contact (+RC).** RC occurs prior to the vertical dashed line. All other descriptions as in Fig 2. Lines correspond to means for groups at: lower-risk (black), higher-risk in the community (blue) and in LTC facilities (red), with shading indicating ± 1 SD. Green shading indicates the shielding phase. NS: no shielding; IS: imperfect shielding; PS: perfect shielding.

## 4: Discussion

Our results demonstrate critical epidemiological weaknesses in shielding strategies that aim to achieve herd immunity by isolating the vulnerable while allowing infections to spread among lower-risk members of the population. While our main results focus on a limited set of parameters, our findings are qualitatively robust to sensitivity analysis (§A of the *S1 Text*). Even in the best-case scenario with perfect shielding, our model estimates that there would have been tens of thousands of avoidable deaths among those deemed to be at lower risk due to limited mitigation in this subpopulation, even without accounting for the rapid depletion of healthcare capacity. A significant reduction in contact rates would have been required to avoid overwhelming healthcare capacity during shielding [18], but the population would have then failed to achieve herd immunity, allowing a second, deadlier wave to occur following the lifting of restrictions. Under more realistic assumptions of imperfect shielding, our model estimates that deaths would have been 150% to 300% higher compared to perfect shielding. Breaking down deaths by risk category and location reveals contrasting effects of the scenarios on different groups (Table B in *S1 Text*). In some cases (+RC), LTC residents have disproportionately higher death rates than similar individuals in the community, and in others the converse is true (IS). This occurs because LTC residents are clustered together within facilities, whereas higher-risk individuals outside of LTC facilities are assumed to mix randomly in the

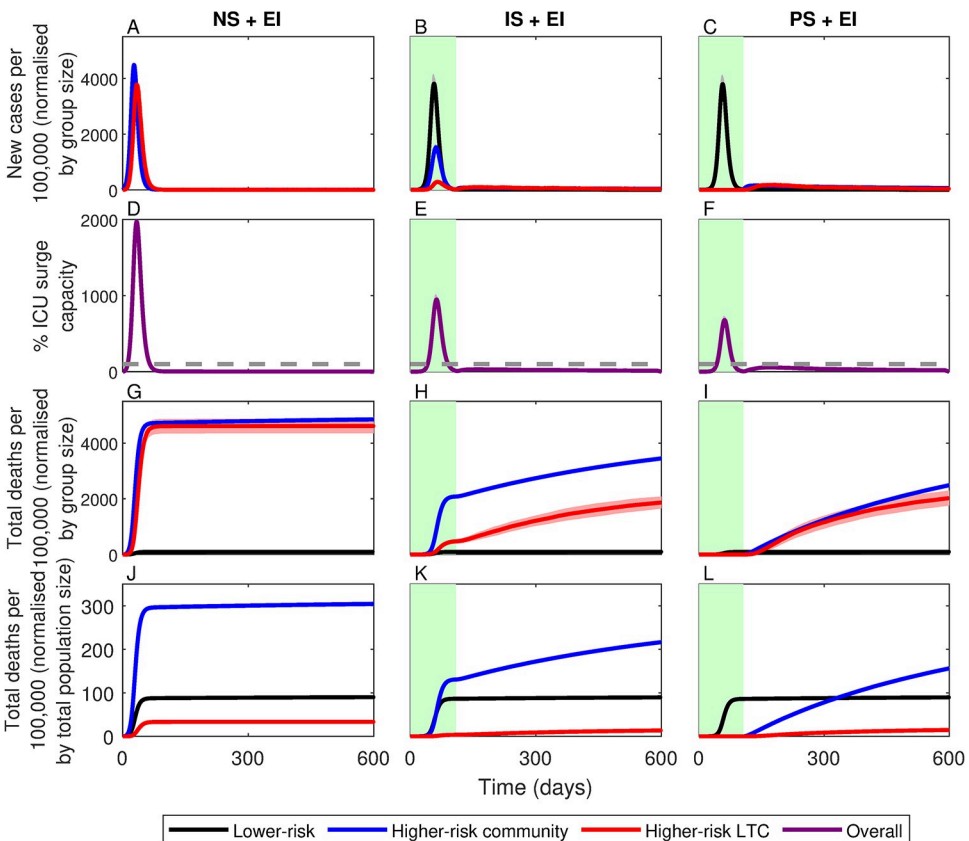

**Fig 5. Simulations of the three shielding scenarios with external infections (+EI).** All other descriptions as in Fig *2*.
Lines correspond to means for groups at: lower-risk (black), higher-risk in the community (blue) and in LTC facilities
(red), with shading indicating ± 1 SD. Green shading indicates the shielding phase. NS: no shielding; IS: imperfect
shielding; PS: perfect shielding.

community. Clustering of susceptible contacts means that higher-risk individuals in LTC facil-
ities are more adversely affected than those in the community when herd immunity is not
reached during the shielding phase, as LTC facilities remain vulnerable to large outbreaks once
restrictions are lifted. This effect is not seen in previous models which do not differentiate
between higher risk members of the population who reside in the community and those who
reside in LTC facilities [6].

Our model demonstrates that shielding would have only worked well under practically
unrealizable conditions. If any of these conditions had not been met, then significant out-
breaks would have occurred in higher-risk subpopulations, leading to many more deaths than
if shielding were perfect. To be effective, shielding would have also required those who were at
higher risk to not only be rapidly and accurately identified, but also to shield themselves for an
indefinite period. If higher-risk individuals were to be misdiagnosed or were unable to fully
isolate this would have decreased the effectiveness of shielding. For example, shielding would
have been especially difficult for households that contained both higher- and lower-risk indi-
viduals (e.g., 74% of CEV people in England live with other people, and 15% live with children
aged under 16 years [29]). The large number of multi-risk households suggests that either
shielding would have been far from perfect, or a significant proportion of lower-risk individu-
als would have also had to shield, in which case it would have been harder (or perhaps impossi-
ble) to achieve herd immunity during the shielding phase.

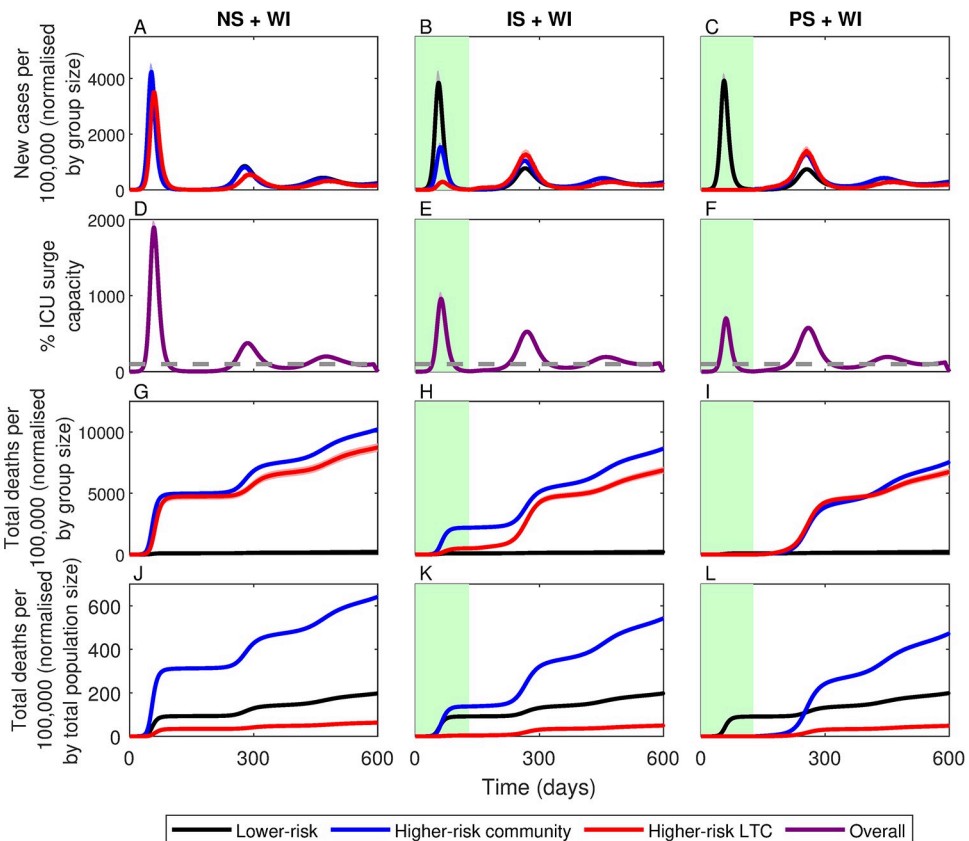

**Fig 6. Simulations of the three shielding scenarios with waning immunity (+WI).** All other descriptions as in Fig 2. Lines correspond to means for groups at: lower-risk (black), higher-risk in the community (blue) and in LTC facilities (red), with shading indicating ± 1 SD. Green shading indicates the shielding phase. NS: no shielding; IS: imperfect shielding; PS: perfect shielding.

The present study focuses on three critical epidemiological weaknesses in shielding strategies, but there are many additional epidemiological, logistical, and ethical problems with shielding that are not captured by our model [9, 30]. Notably, even if perfect shielding had been possible, there would have been major issues associated with the large number of infections required to achieve herd immunity. Long-term sequalae of infection, known collectively as 'long COVID', are thought to affect between 5 and 10% of those infected [31, 32], which would have left many otherwise healthy people with significant long-term health problems. A large epidemic would have also potentially allowed new variants to emerge, which may have been more transmissible, more deadly, or able to escape immunity. We made the conservative assumption of no pathogen evolution, but novel variants would have rendered shielding an even less effective strategy. Our model also made conservative assumptions regarding infection fatality rates (IFRs; see §2.1) and immunity, but more realistic assumptions are likely to make the case for shielding far worse. For example, we used relatively low estimates for the IFRs and assumed that these were fixed even though healthcare capacity would have been significantly overwhelmed under all shielding scenarios. The model also did not capture the impact of healthcare burden on mortality from other causes. We further assumed that immunity from infection was perfect and long-lasting ('best-case' assumptions for shielding), but neither is likely to be true in reality [33]. These additional considerations, in combination with the clear flaws indicated by our modelling, suggest that, while an idealized shielding strategy may have

allowed populations to achieve herd immunity with fewer deaths, they are likely to have failed catastrophically in practice.

## Supporting information

**S1 Text.**
(DOCX)

## Acknowledgments

We thank B. Adams, A. Best, G. Constable, E. Feil, T. Rogers, and R. Thompson for helpful discussions and comments on the manuscript.

## Author Contributions

**Conceptualization:** Christian A. Yates, Ben Ashby.

**Formal analysis:** Cameron A. Smith.

**Investigation:** Cameron A. Smith.

**Methodology:** Cameron A. Smith, Christian A. Yates.

**Visualization:** Cameron A. Smith.

**Writing – original draft:** Cameron A. Smith, Christian A. Yates, Ben Ashby.

**Writing – review & editing:** Cameron A. Smith, Christian A. Yates, Ben Ashby.

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
