## [Decision Letter · Decision Letter 0]

4 Jan 2022

PGPH-D-21-00992

Critical weaknesses in shielding strategies for COVID-19

Dear Dr. Smith,

Thank you for submitting your manuscript to PLOS Global Public Health. After careful consideration, we feel that it has merit but does not fully meet PLOS Global Public Health’s publication criteria as it currently stands. Therefore, we invite you to submit a revised version of the manuscript that addresses the points raised during the review process.

In general, the reports by the two appointed reviewers address important concerns that if appropriately addressed may result in your manuscript considered for publication in PLOS Global Public Health.

We look forward to receiving your revised manuscript.

Kind regards,

Kevin Escandón, MD MSc

Academic Editor

Journal Requirements:

1. We ask that a manuscript source file is provided at Revision. Please upload your manuscript file as a .doc, .docx, .rtf or .tex. If you are providing a .tex file, please upload it under the item type ‘LaTeX Source File’ and leave your .pdf version as the item type ‘Manuscript’.

2. Please provide separate figure files in .tif or .eps format only and remove any figures embedded in your manuscript file.  If you are using LaTeX, you do not need to remove embedded figures.

3. We have noticed that you have uploaded supporting information but you have not included a list of legends.  Please add a full list of legends for all supporting information files (including figures, table and data files) after the references list. 

4. Please note that your Data Availability Statement is currently missing the repository name and/or the DOI/accession number of each dataset OR a direct link to access each database as your provided link cannot be found. If your manuscript is accepted for publication, you will be asked to provide these details on a very short timeline. We therefore suggest that you provide this information now, though we will not hold up the peer review process if you are unable.

5. Please amend your detailed Financial Disclosure statement. This is published with the article, therefore should be completed in full sentences and contain the exact wording you wish to be published.

iii). State what role the funders took in the study. If the funders had no role in your study, please state: “The funders had no role in study design, data collection and analysis, decision to publish, or preparation of the manuscript.”

6. Please make sure that the recepient in the Funding Information matches with the Financial Disclosure statement.

Reviewers' comments:

Reviewer's Responses to Questions

**Comments to the Author**

1. Does this manuscript meet PLOS Global Public Health’s publication criteria? Is the manuscript technically sound, and do the data support the conclusions? The manuscript must describe methodologically and ethically rigorous research with conclusions that are appropriately drawn based on the data presented.

Reviewer #1: Yes

Reviewer #2: Partly

2. Has the statistical analysis been performed appropriately and rigorously?

Reviewer #1: Yes

Reviewer #2: No

3. Have the authors made all data underlying the findings in their manuscript fully available (please refer to the Data Availability Statement at the start of the manuscript PDF file)?

Reviewer #1: No

Reviewer #2: Yes

4. Is the manuscript presented in an intelligible fashion and written in standard English?

Reviewer #1: Yes

Reviewer #2: Yes

5. Review Comments to the Author

Reviewer #1: This paper uses a stochastic SEIRD model to explore the efficacy of shielding and other non-pharmaceutical interventions on a simulated COVID-19 outbreak in a structured population stratified by risk of mortality and location. The study demonstrates that shielding strategies would not effectively protect high-risk individuals in the presence of a realistically structured population across a range of different assumptions regarding the efficacy of shielding, heterogeneity in hospitalisation rates and transmission.

Overall I thought this was a well written paper, with the conclusions supported by the results presented. However, parts of the results section would benefit from being tightened up, so that the conclusions (particularly regarding herd immunity and external infections) are fully supported.

Methods

1. Justification/referencing is needed to clarify the choice of population sizes for small, medium and large LTC facilities (lines 86-90).

2. There is also a need to describe the parameterisation for the transmission pressure from external infections, ηi(t). This is important considering that Fig. 4 is a key result for the paper (line 119).

3. The authors need to explicitely describe parameters pop_l and ICU_l in section 2.3. It is not clear what values in table S3 the equation is referring to.

Results

4. The authors have placed significant emphasis on discussing their results in the context of herd immunity (e.g lines 252-259, 267-268). However, little quantitative exploration of the dynamics of herd immunity or herd immunity thresholds (HIT) has been performed in this study. Supplementary analyses which demonstrate how certain scenarios fail to reach HIT due to insufficient depletion of susceptibles, or reach HIT but still result in excessive number of deaths would be useful to support the statements made in the results section.

5. Hospitalisations exceeding ICU capacity is referenced in the results and plotted on figures, but there is no description in the main text what this ICU capacity or threshold value is (line 224-225, 233-234, 250, 279-280).

6. The authors do not describe the impact of external infections in Fig. 4 sufficiently before contextualising the figure with regards to herd immunity (lines 254-262). The increase in total deaths per 100,000 for IS + EI and PS + EI scenarios is an important result and deserves further explanation. It would also be of interest to conduct an additional sensitivity analysis ranging the extent of ηi(t) and how this affects Fig. 4.

7. It is not clear what the three critical weaknesses are when the authors refer to "A third critical weakness…" (line 267). These weaknesses need to be made more explicit in the text.

8. Sensitivity analyses should be referenced within the main body of text to demonstrate that the results are robust to parameter variation.

9. It is not clear what the authors are suggesting with "Furthermore, behaviour change would likely have been exacerbated if healthcare services were overwhelmed” (line 280-281). Are the authors suggesting that the efficacy of RC interventions would increase if ICU capacity was overwhelmed? This needs further explanation.

Discussion

10. The authors state there would be “tens of thousands of avoidable deaths among those deemed to be lower risk” (line 286). “Avoidable” implies that these deaths are being compared to some baseline. This needs further explanation.

11. The statement "This effect is not seen in previous models which do not account for the clustering of higher risk members of the population", regarding reference (6) is false (lines 300-303) - as both models include population clustering based on risk status (the referenced model has a general, vulnerable and shielder population structure). Do the authors mean heterogeneity across subpopulations with regard to the probability of requiring ICU treatment (hospitalisation-based risk clustering)?

Minor Comments

12. It would be beneficial to briefly explain what "shielding" is in the abstract – as this is only described in the introduction (line 14).

13. Figures 4 and 5 would be more readable with a standalone legend.

14. The probability of requiring ICU treatment given an individual is symptomatic is derived from Table S3 not Table S2 (line 209).

15. Please ensure that MATLAB code is accessible by Github users without a bath.ac.uk login.

Reviewer #2: 

Summary:

This study addresses a key public question and the article will benefit PLoS Global Public Health readers. The study investigates the effectiveness of a ‘shield-strategy’ in which the most vulnerable individuals are shielded against COVID-19 infection. Under the same strategy, the lower risk individuals are allowed to be infected, targeting the attainment of herd immunity. This strategy was among those advocated for at the beginning of the pandemic, yet the findings of this study, even in the best case scenario demonstrate that it would be detrimental to adopt. The paper is generally well written. Besides a few assumption that are unrealistic as indicated as major observations, the methods used are appropriate for the task at hand.

Major observations:

1. You use an SEIR epidemic model and this seems to be overly simplistic given the existing knowledge about COVID-19 dynamics and characteristics. For example, by this simple SEIR formulation, you make no distinction between the asymptomatic and symptomatic infectious individuals. These categories respond differently to the disease (different hospitalization and mortality levels, yet these are key outputs in your study) and also contribute differently to the infection pressure with the symptomatic likely to shed more. Revise your model to further split I into symptomatic and asymptomatic as this will alter the dynamics significantly. You also miss the hospitalized compartment yet its output is among you key results. For an example on how these were captured, see Mugisha et al. 2021 PLoS ONE 16(2): e0247456. https://doi.org/10.1371/journal.pone.0247456 and the cited literature therein)

2. You use and incubation period of 5 days and an infectious period of 2 days. In literature, there is a wide variation in these estimates and it better you explore a wider range for these and perhaps other parameters. Since you are working with a stochastic model, you could deploy a parameter sampling scheme in which the program picks a random number from a given distribution of the parameter at every iteration. The other option would be to perform a sensitivity analysis to assess the impact of these parameters on the study findings. This will improved credibility and reliance of your study findings. You attempt such an analysis by lowering R0 from 3 to 2.5 and also increasing it to 3.5; add further sensitivity analyses to this.

3. Given the existing knowledge, it is clear that suffering from the disease doesn’t confer permanent immunity. Therefore, you need amend you model to allow for waning of disease-induced immunity.

4. In line 145, you write “In the imperfect 145 and perfect shielding scenarios, shielding begins at the start of each simulation and ends once incidence falls below a threshold of 60 new cases per 100,000 in the population per week.” Justify your choice for this criteria.

5. Equation 8f on page 10 is wrong.

Minor observations:

6. The abstract needs to briefly mention the methods used.

7. Keywords are missing.

8. Line 33, change “lower-risk” to lower risk i.e. remove hyphen? Other occurrences of lower-risk are ok.

9. Line 99, you write 75s, write this fully e.g. 75 year-olds for clarity.

10. Line 138, use “element-wise”?

11. Line 177, you simulate 600 days. Isn’t this timeframe long enough to warrant inclusion of the vital dynamics i.e. birth and nature deaths in the model?

6. PLOS authors have the option to publish the peer review history of their article (what does this mean?). If published, this will include your full peer review and any attached files.

**Do you want your identity to be public for this peer review?** For information about this choice, including consent withdrawal, please see our Privacy Policy.

Reviewer #1: **Yes: **Alexander Liang Kang Morgan

Reviewer #2: **Yes: **Amos Ssematimba

---

## [Decision Letter · Decision Letter 1]

24 Feb 2022

PGPH-D-21-00992R1

Critical weaknesses in shielding strategies for COVID-19

Dear Dr. Smith,

Thank you for submitting your manuscript to PLOS Global Public Health. I read your manuscript and reviewers' input on your revised manuscript. I feel that it has substantial merit but there are some minor corrections that are necessary before proceeding to accepting it for publication in PLOS Global Public Health. I invite you to submit a revised version of the manuscript that addresses these points.

If applicable, please ensure that your decision is justified on PLOS Global Public Health’s publication criteria and not, for example, on novelty or perceived impact. Please submit your revised manuscript by Mar 26 2022 11:59PM. If you will need more time than this to complete your revisions, please reply to this message or contact the journal office at globalpubhealth@plos.org. Please include the following items when submitting your revised manuscript:

We look forward to receiving your revised manuscript.

Kind regards,

Kevin Escandón, MD, MSc

Academic Editor

Journal Requirements:

Editor Comments:

This manuscript was substantially improved by following reviewers' advice and addressing most pressing concerns. I agree with authors' rebuttal.

Just some minor final comments:

L47: I think GBD influenced greatly these unfortunate decisions/thinking, rather than partly. Also, I would add, if authors agree, something on the lines of "while allowing uncontrolled viral transmission among 'low-risk' individuals" following focused protection of 'high-risk' individuals. Also, I suggest not citing their website as a way of avoiding potential spread of misinformation and dangerous approaches. Critiques or other type citations may be more appropriate perhaps?

Examples (there may be others preferred by the authors): https://doi.org/10.1136/bmj.m3908
https://theconversation.com/5-failings-of-the-great-barrington-declarations-dangerous-plan-for-covid-19-natural-herd-immunity-148975
https://doi.org/10.1186/s12879-021-06357-4

L66: check this line which is incomplete "(see §2.1 and )"

L123: change "infection with COVID-19" to "SARS-CoV-2 infection"

L308 and L345: Here is the same case as in L66: "descriptions as in" ? The clean version in these three cases seems to drop an error: "Error! Reference source not found." Pleasure ensure citations field codes are not corrupted and citations are accurate throughout the document.

Keywords: These should be consistent between the main text and its supplementary material since it's the same manuscript.

Reviewers' comments:

Reviewer's Responses to Questions

**Comments to the Author**

1. If the authors have adequately addressed your comments raised in a previous round of review and you feel that this manuscript is now acceptable for publication, you may indicate that here to bypass the “Comments to the Author” section, enter your conflict of interest statement in the “Confidential to Editor” section, and submit your "Accept" recommendation.

Reviewer #2: All comments have been addressed

2. Does this manuscript meet PLOS Global Public Health’s publication criteria? Is the manuscript technically sound, and do the data support the conclusions? The manuscript must describe methodologically and ethically rigorous research with conclusions that are appropriately drawn based on the data presented.

Reviewer #2: Yes

3. Has the statistical analysis been performed appropriately and rigorously?

Reviewer #2: Yes

4. Have the authors made all data underlying the findings in their manuscript fully available (please refer to the Data Availability Statement at the start of the manuscript PDF file)?

Reviewer #2: No

5. Is the manuscript presented in an intelligible fashion and written in standard English?

Reviewer #2: Yes

6. Review Comments to the Author

Reviewer #2: Well done on the rebuttal, I appreciate that you performed extra analyses and I also understand that some of the suggests edits could make the model complex to analysis and may mask the key findings of the study. With the extra analyses included, the manuscript has been improved upon.

7. PLOS authors have the option to publish the peer review history of their article (what does this mean?). If published, this will include your full peer review and any attached files.

**Do you want your identity to be public for this peer review?** For information about this choice, including consent withdrawal, please see our Privacy Policy.

Reviewer #2: **Yes: **Amos Ssematimba

---

## [Editor Report · Decision Letter 2]

6 Mar 2022

Critical weaknesses in shielding strategies for COVID-19

PGPH-D-21-00992R2

Dear Dr. Smith,

We are pleased to inform you that your manuscript 'Critical weaknesses in shielding strategies for COVID-19' has been provisionally accepted for publication in PLOS Global Public Health.

Best regards,

Kevin Escandón, MD, MSc

Academic Editor